# Development of a core outcome set to determine the overall treatment success of acute uncomplicated appendicitis in children: a study protocol

Frances C Sherratt,[1] Simon Eaton,[2] Erin Walker,[3] Lucy Beasant,[4] Jane M Blazeby,[5] Bridget Young,[1] Esther Crawley,[4] Wendy W Wood,[6] Nigel J Hall[7]

► Prepublication history and additional material is available. To view please visit the journal (http://dx.doi.org/ 10.1136/bmjpo-2017-000151).

For numbered affiliations see end of article.

**Correspondence to**
Dr Frances C Sherratt; sherratt@liverpool.ac.uk

## ABSTRACT

**Introduction** In recent years, there has been growing interest in alternatives to appendicectomy. In particular, non-operative treatment of appendicitis, with antibiotics alone, has been proposed as a potential treatment. A small number of randomised controlled trials (RCTs) in adults and, more recently, children suggest that antibiotic treatment may be a valid alternative to appendicectomy. However, there is currently insufficient data to justify its widespread use. Prior to performing further efficacy studies of the treatment of appendicitis in children, it is imperative to identify the most relevant outcome measures for inclusion in the design of comparative studies. This is of particular importance when evaluating a novel treatment approach since the outcomes of importance may differ from those commonly reported with traditional therapies. A review of the relevant literature and electronic resources failed to identify a core outcome set (COS) for children with appendicitis. We aim to define a COS for the measurement of treatment interventions in children (<18 years) with acute appendicitis.

**Methods and analysis** This project will entail: (1) a systematic review to identify previously reported acute uncomplicated appendicitis treatment outcomes; (2) assembly of stakeholder panels (paediatric and adult surgeons, patients and parents); (3) a three-stage Delphi process; and (4) a final consensus meeting to complete the COS.

**Ethics and registration** COS development is part of CONservative TReatment of Appendicitis in Children - a randomised controlled Trial (Feasibility) (CONTRACT) study, for which full ethical approval for CONTRACT has been granted. The COS development study is registered with the COMET Initiative in May 2017 (http://www.comet-initiative.org/studies/details/987).

## What is already known on this topic?

► Traditionally, appendicectomy has been the gold standard treatment for acute appendicitis in children, but there has been increased interest in non-operative treatment (with antibiotics).
► Core outcome sets are developed and adopted to avoid inconsistencies in outcome selection, measurement and reporting that may otherwise exist.
► There is currently no core outcome set for the measurement of effectiveness of treatment interventions in children with acute uncomplicated appendicitis.

## What this study hopes to add?

► This project will involve defining a core outcome set for the measurement of effectiveness of treatment interventions in children with acute uncomplicated appendicitis.
► Considering outcomes of importance to patients, parents of patients and health professionals is crucial for paediatric appendicitis research to be meaningful and relevant.

## BACKGROUND

A lack of knowledge and understanding regarding which outcomes are important to patients and clinicians may result in important outcomes being omitted from clinical trials. Differences in outcome selection and reporting between studies and how outcomes are defined and measured also make it difficult, sometimes impossible, to synthesise results of studies (eg, meta-analysis) and apply them in a meaningful way. To address these problems, core outcome sets (COS) have been proposed as a means of standardising outcome selection, measurement and reporting in healthcare research and in clinical trials in particular.[1 2] The development of a COS and its adoption by researchers is intended to help avoid inconsistencies in outcome selection, measurement and reporting that may otherwise exist. If trials do not adopt an established COS, they risk selecting suboptimal outcomes and are unlikely to contribute usable information.[3]

For children with acute uncomplicated appendicitis, appendicectomy has traditionally been the gold standard treatment, but there has been growing interest in alternatives to appendicectomy. In particular, non-operative treatment of appendicitis, with antibiotics, has been proposed as a potential treatment. A small number of randomised controlled trials (RCTs) in adults[4–6] and, more recently, RCTs and prospective preference studies in children[7 8] suggest that antibiotic treatment may be a valid alternative to appendicectomy.[9] However, there is currently insufficient data to justify its widespread use. Prior to performing further efficacy studies of the treatment of appendicitis in children, it is imperative to identify the most relevant outcomes for inclusion in the design of comparative studies. This is of particular importance when evaluating a novel treatment approach since the outcomes of importance may differ from those commonly reported with traditional therapies. A review of the relevant literature and electronic resources failed to identify a COS for children with appendicitis.[10] Furthermore, a wide range of outcomes were reported, and a range of different primary outcomes were used across studies.

In order to advance our understanding of which outcomes are important and to fulfil an unmet need in our future research programme, the aim of this study is to develop a COS for the measurement of effectiveness of treatment interventions in children (<18 years) with acute uncomplicated appendicitis.

## METHODS

The COS development is a component of a wider project, CONservative TReatment of Appendicitis in Children - a randomised controlled Trial (Feasibility) (CONTRACT) study (http://www.nets.nihr.ac.uk/projects/hta/1419290). The COS development study was registered with the COMET Initiative in May 2017 (http://www.comet-initiative.org/studies/details/987).

### Scope of the COS

The COS is intended to be used to evaluate the overall success of operative and non-operative treatment among children who are assigned a clinical and/or radiological diagnosis of acute uncomplicated appendicitis. The COS will include outcome measures identified as important within 12 months of treatment initiation and longer term outcomes if applicable. The COS focuses specifically on treatment of acute uncomplicated appendicitis; the treatment of perforated appendicitis (with or without abscess) and appendix mass is outside the scope of this COS. The key objectives of the study are:

1. to determine which outcomes have previously been reported in studies comparing treatments for acute uncomplicated appendicitis in children
2. to prioritise treatment outcomes of children with acute uncomplicated appendicitis from key stakeholder groups' perspectives (including paediatric sur-

geons, general surgeons, patients (12–18 years old) and parents of children who have had acute uncomplicated appendicitis)
3. to compare and contrast paediatric acute uncomplicated appendicitis treatment outcomes prioritised by key stakeholder groups (as detailed above)
4. to achieve consensus between key stakeholder groups on a COS to evaluate overall success of treatment for acute uncomplicated appendicitis in children.

### Design

COS development will entail four key stages:

1. a systematic review to identify previously reported acute uncomplicated appendicitis treatment outcomes
2. assembly of stakeholder panels
3. a three-phase online Delphi process
4. consensus meeting.

The CONTRACT study, as part of the feasibility study, will involve in-depth qualitative research. Interviews will be conducted with patients, parents and caregivers to explore experiences of treatment for appendicitis. We will explore families' perceptions of meaningful and important outcomes, which will help to optimise COS participants' project understanding and engagement throughout the COS development.

### Systematic review

The COMET Initiative recommend the use of systematic reviews in informing the first phase of the Delphi process.[11] Two recent systematic reviews will be used to inform the initial list of potential outcomes to be considered for the COS. The first review identified outcomes used in studies of paediatric appendicitis,[10] and the second review aimed to determine safety and efficacy of non-operative treatment for acute appendicitis.[9] Identified outcomes were combined with closely similar outcomes from the operative treatment systematic review[10] to develop an initial list of outcomes (online supplementary appendix 1). Online supplementary appendix 1 describes the eligibility criteria for inclusion of papers. All relevant articles will therefore be included that reported any non-operative treatment regimen for acute uncomplicated appendicitis in children with or without a comparative group of children undergoing surgical treatment. All outcomes identified through these reviews and an updated literature search will inform the initial list of outcomes and will be considered by the stakeholder groups as part of the Delphi process.

### Finalising and appropriate wording of initial outcomes

To inform and support the CONTRACT study, a Study Specific Advisory Group (SSAG) has been assembled, comprising 15–20 young people and parents. Young people recruited are children who have had appendicitis or children from the existing Clinical Research Network (Children) Young Persons Advisory Groups. Parents are parents of children who have had appendicitis. An SSAG

**Table 1** Core outcome set stakeholder groups and methods of approaching potential participants

| Stakeholder group | Selection criteria | Method of approach |
|---|---|---|
| Patients and parents | ► Patients aged 12–18 years who have been treated for acute uncomplicated appendicitis in the preceding 12–24 months.<br>► Parents of children (aged 5–18 years) who have been treated for acute uncomplicated appendicitis in the preceding 12–24 months.<br>► Families may or may not have participated in CONTRACT.<br>► Patient and parent panels will specifically include children and parents treated initially by non-operative management as well as those treated operatively. | ► Invited to participate via clinical teams from the three sites that are participating in the CONTRACT study.<br>► Identified to participate via further participant identification sites. |
| Paediatric surgeons | ► All practising consultant paediatric surgeons in the UK who treat children with acute uncomplicated appendicitis will be considered potential participants. | ► Invited to participate via the mailing list of the British Association of Paediatric Surgeons and through personal contacts of the investigators. |
| General surgeons | ► Adult general surgeons in the UK who regularly treat children with acute uncomplicated appendicitis will be considered potential participants. This will include those identified as having an interest in the treatment of children. | ► Invited to participate via existing personal contacts and through regional paediatric surgical networks within the UK. |

CONTRACT, CONservative TReatment of Appendicitis in Children—a randomised controlled Trial (Feasibility).

meeting will be held to present the initial list of treatment outcomes, to inform the addition and wording of outcomes and to ensure outcomes are appropriately presented. Three versions of the stakeholder-facing materials will be developed for all rounds of the Delphi process, for each stakeholder panel (clinicians, young people and parents), using appropriate language identified and agreed by the SSAG. The CONTRACT qualitative study will also inform the initial outcomes.

**Stakeholder panel assembly: identification and recruitment**

For the COS to be meaningful and relevant to those involved in the treatment of acute appendicitis, the COS needs to reflect the views of patients who have been treated for acute appendicitis, their parents and relevant clinicians. As these groups may have different priorities that could obstruct reaching consensus on a COS, the stakeholders will be separated into three panels, which we intend to be equally weighted: (1) patients; (2) parents; and (3) paediatric surgeons and general surgeons. Initially, potential members of each stakeholder panel known to the research team will be invited to participate, and we will develop strategies to identify further experts (see table 1).

Initial contact with potential participants will explain the study and why they have been identified as a potential participants. It will contain a plain language summary of the study aims and procedures, emphasising the importance of commitment to the panel. The wording of this initial contact will be tailored to meet the panel category. It will also contain a link to an online form to enable participants to express their interest in participation in the study and to provide further information on their experience of the treatment of acute appendicitis. We will

ask participants to commit to completing three rounds of questionnaires anticipated to take approximately 10 min each to complete.

The process of invitation and enrolment will continue until the optimal number of stakeholders have expressed an interest to participate (with at least 10 in each panel). There is no consensus on the optimal sample size for a Delphi study; recruitment will therefore be based on previous Delphi studies.[12] We will aim to achieve 75–100 participants in the first round of the Delphi with at least as many parents/children as clinicians. Efforts will be made to invite a diverse range of participants to each stakeholder group. We will aim to send the first questionnaire to all participants on the same day they each confirm their desire to participate.[13] Participants will be sent a link to a customised online database hosted on a secure server, from which they can access and complete phase one questionnaire of the Delphi process. To limit attrition, appropriate procedures will be completed,[12 14] including reminder emails.

Participation in this COS development process will be limited to surgeons, patients and parents from the UK. We will not recruit paediatric or general surgeons from outside the UK because the treatment pathway for appendicitis in the UK differs to that in other countries. Furthermore, if we were to recruit surgeons from outside the UK, we would also need to recruit patients and parents from outside the UK to avoid bias; this would become increasingly challenging logistically.

**Definition of consensus**

During the process, participants will be asked to score each outcome using the Grading of Recommendations, Assessment, Development and Evaluations (GRADE)

scale, which is recommended by the COMET Initiative.[15] The scale will be presented in the format 1–9, with 1–3 labelled 'not important', 4–6 labelled 'important but not critical' and 7–9 labelled 'critical'.[16] 'Consensus in' will be defined as ≥70% of participants rating the outcome 7–9 and <15% rating it as 1–3. Outcomes will defined as 'consensus out' if >70% participants rate it 1–3 and <15% rate it 7–9. Outcomes not meeting these definitions will be classified as 'no consensus'.

### Delphi process: phase one data collection
A customised online data system (developed by the National Perinatal Epidemiology Unit (University of Oxford, UK)) will be used to conduct a three-phase Delphi process run in parallel across the stakeholder panels. Participants will be presented with the initial list of outcomes, grouped by domains. As described, the initial list will comprise outcomes identified in our recent systematic review,[10] updated review of the literature (online supplementary appendix 1) and any additional outcomes identified during qualitative interviews with key stakeholders. There will also be an option for experts to add further outcomes, but these outcomes will not be scored in phase one.

Surgeon participants will be asked the key question 'How important do you consider the following outcomes to be when considering which treatment to offer children with uncomplicated acute appendicitis?' A similar question will be posed to patient and parent stakeholder panels, with the wording altered as necessary based on our SSAG input.

Participants will be asked to complete each round of the Delphi exercise within 3 weeks, and two subsequent reminder emails will be sent. Participants who have not completed the questionnaire within 4 weeks of being requested to complete the questionnaire will be deemed not to have completed that phase.

### Delphi process: phase one data analysis
The number of participants who were invited to participate in phase one and the response rate from each stakeholder group will be recorded. Outcomes will be analysed separately for each panel, and descriptive statistics will be calculated. All outcomes will be carried forward to phase two.

Additional outcomes provided by participants will be reviewed by two members of the COS team to ensure they represent new outcomes and will be included in phase two if they were proposed by at least two participants.

### Delphi process: phase two data collection
Participants who completed phase one will be invited to participate in phase two. Participants will be individually presented with their own scores, the distribution of scores for each outcome from their stakeholder group in phase one, and will be asked to rescore each outcome. Participants will be asked to score any new outcomes identified in phase one.

### Delphi process: phase two data analysis
The data analysis process described for phase one will be repeated. Bias from loss of experts between phases will be assessed. Any outcomes that meet the criteria of 'consensus out' will be removed from the outcomes list prior to phase three. All other outcomes from phase two will be carried forward to phase three.

### Delphi process: phase three data collection
Participants who completed phases one and two will be invited to participate in phase three. The data collection process described for phase two will be repeated; however, participants will also be shown scores for their own stakeholder panel and separately for each other panel. This will allow participants to consider other stakeholder groups' views before rescoring the outcomes.[17] Participants will be asked to identify the one outcome which they believe is the most important for informing their treatment choice, and if they cannot identify a single outcome, a combination of essential outcomes.

### Delphi process: phase three data analysis
The data analysis process described for phase two will be repeated. All outcomes from phase three will be carried forward to the consensus meeting.

### Final consensus meeting
The aim of the consensus meeting is to ratify outcomes where consensus ('in' or 'out') has been achieved, to discuss outcomes where consensus could not be achieved and to finalise the COS. All participants who have completed all three rounds of the Delphi exercise will be invited to attend the consensus meeting. We will aim to have a minimum of 40 stakeholders confirm their attendance with equally weighted panels and disciplines. Representatives from each stakeholder panel will be required in order for the consensus meeting to be quorate.

During the meeting, stakeholders will be provided with an overview of the results of phase three including presentation of each outcome scored, how it was scored by each stakeholder panel and its consensus status. Following moderated discussion, each outcome will be anonymously rescored using the same scoring system as the Delphi process. For outcomes for which 'no consensus' was achieved across all stakeholder panels at the end of the Delphi exercise, and for which consensus was achieved in at least one but not all stakeholder groups, further discussion will take place, following which attendees will be asked to score each outcome anonymously. Following rescoring at the consensus meeting, outcomes reaching 'consensus in' will be included in the finalised COS. All others will be excluded.

### Finalising the COS
The Outcome Measures in Rheumatology initiative provides a COS development framework that is a useful across various healthcare domains.[18] In developing the current COS, we will draw on this framework, which

comprises three core domains (death, life impact and pathophysiological manifestations) and one strongly recommended domain (resource use). The framework recommends inclusion of at least one applicable measurement instrument for each core domain. It also recommends inclusion of 'adverse events'. Overall, we aim to achieve a manageable COS with a maximum of approximately 10 outcomes. Beyond the current project, further work may be undertaken to establish optimal methods of measuring each core outcome[18] and subsequent work may be undertaken to determine whether the UK-based COS is valid internationally.

## Data management

Experts will enter data directly into the customised database when they complete each questionnaire at each phase of the Delphi process. Anonymised data will be stored securely and will be managed as per standard operating procedures. Only selected members of the research team will have access to the data.

## Ethics and dissemination

Full ethical approval for CONTRACT was granted by South Central - Hampshire A Research Ethics Committee (REC ref: 16/SC/0596) in November 2016. The finalised COS will be disseminated via publication, events and the COMET Initiative website (http://www.comet-initiative.org/).

**Author affiliations**
[1]Department of Psychological Sciences, Institute of Psychology, Health & Society, University of Liverpool, Liverpool, UK
[2]Developmental Biology and Cancer Programme, UCL Great Ormond Street Institute of Child Health, London, UK
[3]Great Ormond Street Hospital NHS Foundation Trust, London, London, UK
[4]Centre for Child and Adolescent Health, School of Social and Community Medicine, University of Bristol, Bristol, UK
[5]Centre for Surgical Research, School of Social & Community Medicine, University of Bristol, Bristol, UK
[6]Research Design Services South Central, University of Southampton, Southampton, UK
[7]University Surgery Unit, Faculty of Medicine, University of Southampton, Southampton, UK

**Acknowledgements** All authors gratefully acknowledge funding from the National Institute for Health Research Health Technology Assessment Programme—the CONTRACT trial, including the COS Development. SE gratefully acknowledges funding from Great Ormond Street Children's Charity.

**Contributors** FCS, SE and NJH developed a first draft of the protocol. All authors advised on the development of the protocol and contributed towards the revision of the protocol for final submission.

**Funding** The CONTRACT study is supported by the United Kingdom National Institute for Health Research Health Technology Assessment Programme (Grant number: 14/192/90 http://www.nets.nihr.ac.uk/projects/hta/1419290).

**Competing interests** None declared.

**Ethics approval** South Central—Hampshire A Research Ethics Committee (REC ref: 16/SC/0596).

**Provenance and peer review** Not commissioned; externally peer reviewed.

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
