## [Reviewer comments · BMJ Paediatrics Open]

ARTICLE DETAILS

TITLE (PROVISIONAL)	Development of a core outcome set to determine the overall treatment success of acute uncomplicated appendicitis in children: a study protocol
AUTHORS	Sherratt, Frances; Eaton, Simon; Walker, Erin; Beasant, Lucy; Blazeby, Jane; Young, Bridget; Crawley, Esther; Wood, Wendy; Hall, Nigel

VERSION 1 - REVIEW

REVIEWER	Offringa, Martin The Hospital for Sick Children Toronto, Canada Competing interests: None
REVIEW RETURNED	19-Jun-2017

GENERAL COMMENTS	This article describes the methods for developing a core outcome set (COS) for acute uncomplicated appendicitis in children. Methods include a systematic literature review and consensus process (Delphi plus face to face (?) meeting) following published standards for COS development. It is an important and timely topic, a well-written manuscript by a very strong team of experts in the field. We have some queries that may improve the paper: 1. Please explicitly define acute appendicitis to ensure the health condition that this COS applies to is clear to all end-users and stakeholders;2. It is unclear how the general surgeons will be identified and approached for participation;3. Can the authors provide a justification for only including surgeons in the UK? How will this impact the generalizability and implementation of the COS globally?;4. The article mentions inclusion of studies in children from 12-18 years. Yet, the age range on the COMET registration of this project reports including children 5-18 years. Is there a reason to exclude (outcomes in) children 5-12 years? If so, can the authors provide a justification for this exclusion?;5. Can the authors clarify: will all types of outcomes (i.e. lab biomarkers, clinical, PRO/QoL) be included for consideration for the COS?;6. Finalising the COS. The paper focusses on WHAT to measure as an outcome. The refereed OMERACT filter method to develop a COS (ref #19) also describes the second part: HOW to measure the outcomes. For which outcomes in the provided tables do the authors foresee that there will be issues around the measurement of the outcome in children, and how will the get from a Core Domain Set to a Core Outcome Measurement Set (ref #19)?;
--

	7. Is there an online system for data capture like REDCap, Delphi Manager?; 8. It is stated on page 4 that “the CONTRACT study, as part of the feasibility study, will involve in-depth qualitative research with patients, parents, and caregivers. This qualitative research will be used at each stage of the process to inform project development”. It would be helpful to provide some information on the type of qualitative information to be collected, how it will be collected, analyzed, and how it will be used in the context of this COS development; 9. Are there any processes in place to ensure adequate representation of diverse viewpoints within the stakeholders’ groups (e.g., balanced gender representation, inclusion of different ethnicities, various surgical “schools of thought”)? ; 10. The authors note that representatives from each stakeholder panel must be part of the final consensus meeting. Is there a minimum number of participants from each stakeholder group?; 11. The COS would probably benefit from adding additional stakeholders’ groups, such as policy makers, nurses/nurse practitioners, and researchers who will be end users of the COS for evidence synthesis (SRs/CPGs) and decision making. Please discuss why which stakeholder groups were (not) added.
--	--

REVIEWER	Lee, Steven Harbor-UCLA, USA Competing interests: None
REVIEW RETURNED	06-Jul-2017

GENERAL COMMENTS	Well-written manuscript. The authors address an important need for addressing optimal outcomes for non-operative treatment of appendicitis. The methods are clear and consistent with previous studies. Please correct the headings for table 1.
--

VERSION 1 – AUTHOR RESPONSE

REVIEWER 1

COMMENT 1. PLEASE EXPLICITLY DEFINE ACUTE APPENDICITIS TO ENSURE THE HEALTH CONDITION THAT THIS COS APPLIES TO IS CLEAR TO ALL END-USERS AND STAKEHOLDERS.

Response:

The COS is intended for children who are assigned a clinical and/or radiological diagnosis of acute uncomplicated appendicitis. We have further emphasised this under the heading “Scope of the COS” i.e. “including in future research evaluating treatments (of all modalities) of children who are assigned a clinical and/or radiological diagnosis of acute uncomplicated appendicitis.”

COMMENT 2. IT IS UNCLEAR HOW THE GENERAL SURGEONS WILL BE IDENTIFIED AND APPROACHED FOR PARTICIPATION.

Response: General surgeons will be invited through existing personal contacts and through regional paediatric surgical networks within the UK. This has been highlighted further in Table 1.

COMMENT 3. CAN THE AUTHORS PROVIDE A JUSTIFICATION FOR ONLY INCLUDING SURGEONS IN THE UK? HOW WILL THIS IMPACT THE GENERALIZABILITY AND IMPLEMENTATION OF THE COS GLOBALLY?

Response: The COS development is funded for a consensus meeting in the UK. In general, the treatment pathway for a patient with appendicitis in the UK is somewhat different to that in other countries, which is one of the reasons to conduct a feasibility study (CONTRACT trial) in the UK even though other feasibility and safety studies have been conducted internationally. We believe that it is particularly important that patients and parents are represented at each stage of the COS development, including the consensus meeting. Although it would be possible to include international patients and parents in the online COS phases, for logistical reasons it would not be possible to include these stakeholder groups in the consensus meeting, and we therefore took the decision that to include international participants in the other stakeholder groups would have introduced a potential bias. We are therefore explicit that the COS is developed in the UK. We then envisage a subsequent piece of work to determine whether the UK-based COS is valid internationally. This has been highlighted further in Table 1 and mentioned under the header of "Finalising the COS".

COMMENT 4. THE ARTICLE MENTIONS INCLUSION OF STUDIES IN CHILDREN FROM 12-18 YEARS. YET, THE AGE RANGE ON THE COMET REGISTRATION OF THIS PROJECT REPORTS INCLUDING CHILDREN 5-18 YEARS. IS THERE A REASON TO EXCLUDE (OUTCOMES IN) CHILDREN 5-12 YEARS? IF SO, CAN THE AUTHORS PROVIDE A JUSTIFICATION FOR THIS EXCLUSION?

Response: The target age range of the COS is children 5-18 years, hence the COMET registration. Although the parent stakeholder group will include parents of any age of child with acute appendicitis, it was felt that it would be extremely realistically difficult to include children younger than 12 as a truly independent stakeholder group, given time demands, understanding of the process, consensus meeting etc. We therefore have restricted the patient group to 12-18 years but all other aspects of the COS are 5-18 years. We have emphasised this further in Table 1.

COMMENT 5. CAN THE AUTHORS CLARIFY: WILL ALL TYPES OF OUTCOMES (I.E. LAB BIOMARKERS, CLINICAL, PRO/QOL) BE INCLUDED FOR CONSIDERATION FOR THE COS?

Response: All aspects that were identified by the current and previous systematic review are included as outcomes that will be considered by the stakeholder groups as part of the Delphi process. This is now emphasised under the header "Systematic review".

COMMENT 6. FINALISING THE COS. THE PAPER FOCUSSES ON WHAT TO MEASURE AS AN OUTCOME. THE REFEREED OMERACT FILTER METHOD TO DEVELOP A COS (REF #19) ALSO DESCRIBES THE SECOND PART: HOW TO MEASURE THE OUTCOMES. FOR WHICH OUTCOMES IN THE PROVIDED TABLES DO THE AUTHORS FORESEE THAT THERE WILL BE ISSUES AROUND THE MEASUREMENT OF THE OUTCOME IN CHILDREN, AND HOW WILL THE GET FROM A CORE DOMAIN SET TO A CORE OUTCOME MEASUREMENT SET (REF #19)?

Response: The scope of the current project involves development of the COS. Further work will be needed beyond the current project to establish how best to measure the outcomes (which will likely involve further meetings). Beneath the header "Finalising the COS" this is acknowledged.

COMMENT 7. IS THERE AN ONLINE SYSTEM FOR DATA CAPTURE LIKE REDCAP, DELPHI MANAGER?

Response: The Delphi process will be undertaken using an online data system developed by the National Perinatal Epidemiology Unit (University of Oxford, UK) for use in COS development. This is detailed beneath the header "Delphi Process: phase one data collection".

COMMENT 8. IT IS STATED ON PAGE 4 THAT "THE CONTRACT STUDY, AS PART OF THE FEASIBILITY STUDY, WILL INVOLVE IN-DEPTH QUALITATIVE RESEARCH WITH PATIENTS, PARENTS, AND CAREGIVERS. THIS QUALITATIVE RESEARCH WILL BE USED AT EACH STAGE OF THE PROCESS TO INFORM PROJECT DEVELOPMENT". IT WOULD BE HELPFUL TO PROVIDE SOME INFORMATION ON THE TYPE OF QUALITATIVE INFORMATION

TO BE COLLECTED, HOW IT WILL BE COLLECTED, ANALYZED, AND HOW IT WILL BE USED IN THE CONTEXT OF THIS COS DEVELOPMENT.

Response: Interviews will be conducted to explore patients', parents' and caregivers' experiences of treatment for acute non-complicated appendicitis. We will explore participants' perceptions of meaningful and important outcomes, which will feed in to the COS development process. We have elaborated on this beneath the header, "Design".

COMMENT 9. ARE THERE ANY PROCESSES IN PLACE TO ENSURE ADEQUATE REPRESENTATION OF DIVERSE VIEWPOINTS WITHIN THE STAKEHOLDERS' GROUPS (E.G., BALANCED GENDER REPRESENTATION, INCLUSION OF DIFFERENT ETHNICITIES, VARIOUS SURGICAL "SCHOOLS OF THOUGHT")?

Response: There are no formal processes in place to ensure diversity, although we will informally attempt to invite a diverse range of participants to each stakeholder group. We have emphasised this under the header, "Stakeholder panel assembly - identification and recruitment".

COMMENT 10. THE AUTHORS NOTE THAT REPRESENTATIVES FROM EACH STAKEHOLDER PANEL MUST BE PART OF THE FINAL CONSENSUS MEETING. IS THERE A MINIMUM NUMBER OF PARTICIPANTS FROM EACH STAKEHOLDER GROUP?

Response: In protocol we stated that we aim to have minimum of 40 overall with representation from each group at the consensus meeting. So we do not have a minimum number but we will aim for equal representation from each stakeholder panel. This is detailed beneath the heading, "Final consensus meeting".

COMMENT 11. THE COS WOULD PROBABLY BENEFIT FROM ADDING ADDITIONAL STAKEHOLDERS' GROUPS, SUCH AS POLICY MAKERS, NURSES/NURSE PRACTITIONERS, AND RESEARCHERS WHO WILL BE END USERS OF THE COS FOR EVIDENCE SYNTHESIS (SRS/CPGS) AND DECISION MAKING. PLEASE DISCUSS WHY WHICH STAKEHOLDER GROUPS WERE (NOT) ADDED.

Response: We believe that decision-making around the treatment of acute appendicitis in children is, and will be, made by surgeons, parents and children. In particular, uptake of conservative treatment of acute appendicitis in children will depend on patient, parent and surgeon perception of the merits and demerits of this mode of treatment, and ultimately a consensus of the way to treat an individual child, and this COS is therefore based on these stakeholder groups. Although the other stakeholder groups mentioned by the reviewer may have other views, the team felt that inclusion of other stakeholder groups might dilute the views of those most invested in the decision making process.

REVIEWER 2

COMMENT 1. PLEASE CORRECT THE HEADINGS FOR TABLE 1.

Response: The table has been reformatted to reflect the headers more efficiently. Please see Table 1.

Finally, it has been necessary to make small changes to reduce the word count due to the addition of content, however, no key details have been omitted. I trust that we have provided satisfactory answers to all comments by the reviewers and look forward to your response.